# Post-Pandemic Burden of COVID-19-Related Restrictions in the Management of Digestive Tract Cancers: A Single Center Study

**DOI:** 10.3390/healthcare12060691

**Published:** 2024-03-19

**Authors:** Andreea-Luiza Palamaru, Gheorghe G. Balan, Gabriela Stefanescu, Diana Dumitrascu, Elena Toader

**Affiliations:** 1Faculty of Medicine, “Grigore T. Popa” University of Medicine and Pharmacy Iasi, 700101 Iasi, Romania; luiza.palamaru@yahoo.com (A.-L.P.); gabriela.stefanescu@gmail.com (G.S.); toader.elena@yahoo.com (E.T.); 2Institute of Gastroenterology and Hepatology, 700101 Iasi, Romania; 3Radiology Department, “St. Spiridon” Emergency County Hospital, 700101 Iasi, Romania; ddianalacramioara@gmail.com

**Keywords:** COVID-19, endoscopy, gastrointestinal cancers, delay, pandemic, ethics

## Abstract

The COVID-19 pandemic caused by severe acute respiratory syndrome coronavirus 2 (SARS-CoV-2) infection has required a complete change in the management of patients with gastrointestinal disease who needed to undergo endoscopic procedures. In the second year of the COVID-19 pandemic, due to restrictions for elective endoscopic procedures, a large number of cancer patients were prevented from early diagnosis of several digestive cancers, which has led to a serious burden in the health system that now needs to be dealt with. We designed a prospective study that included patients in whom access to elective endoscopic examinations during the COVID-19 pandemic had been delayed. Our aim was to investigate the impact of the COVID-19 pandemic on the diagnosis rate of digestive tract malignancies in the context of health crisis management that generates an ethical dilemma regarding the balance of utilitarianism versus deontology. Our study shows that the decrease in the number of newly diagnosed gastrointestinal cancers by endoscopy and biopsy during the pandemic restrictions and the delay in diagnosis have had a clear impact on stage migration due to disease progression.

## 1. Introduction

During the Corona Virus Disease-19 (COVID-19) pandemic, more than 35 million people were infected with the SARS-CoV-2 virus and more than 1 million deaths were recorded. Despite the fact that in Romania ischemic heart disease was the main cause of mortality, in 2020, COVID-19 caused approximately 16,000 deaths in Romania (5% of all deaths). However, the indicator of excess mortality suggests that the number of direct and indirect deaths caused by COVID-19 in 2020 could be considered much higher. Several preventive measures must be taken to avoid the spread of infection among healthcare professionals and patients with digestive disease, including the use of personal protective equipment, greater attention to endoscopic room hygiene and rescheduling of non-urgent procedures. The COVID-19 pandemic has had a significant impact on access to care for cancer patients [1]. This pandemic can affect the economy and cause social and political disruptions. This rise of the pandemic can be attributed to global travel and the exploitation of the environment. For these outbreaks to subside and be prevented, there is an urgent need to identify emerging outbreaks and create policies to act accordingly. Well-planned public health structures, such as the Centers for Disease Control and Prevention (CDC), and societal policies are needed to disseminate public health preparedness and guide the emergent response, as well as identify gaps in knowledge and solve them [2]. The prevention of transmission and the treatment of patients with SARS-CoV-2 infection were the main objectives of doctors, which affected other programs in the health systems, such as the diagnosis and treatment of oncological diseases [3]. COVID-19 is a threat to patients with chronic conditions, including those with malignancies [4]. Awareness of the need to prioritize the provision of medical care represented one of the challenges of the COVID-19 pandemic. Reorganization and diminishing the current activities were imposed, with the preservation of urgent procedures and the postponement of semi-urgent and/or elective procedures. The pandemic shows its consequences not only through the number of deaths or patients with pulmonary sequelae but also through an important segment of patients who presented symptoms of the upper or lower digestive tract but who, due to the decrease in the elective endoscopic examinations, did not benefit from timely diagnosis [5]. The fear of infection with the SARS-CoV-2 virus caused a decrease in the addressability of patients so that a large number of gastrointestinal cancers remained undiagnosed or untreated. The prognosis of patients with gastrointestinal malignancies is profoundly affected if the standard care is delayed. Globally, the health policies imposed by the WHO have generated the impossibility of treating all patients, which has determined an ethical dilemma pertaining to the balance of utilitarianism versus deontology regarding the patient’s access to public health services.

In 2020, the year of the COVID-19 pandemic, there were numerous restrictions and legislative regulations that affected the diagnosis and treatment of cancers. Access to investigations and treatments was restricted by the policies imposed by the WHO. Since the outbreak of the COVID-19 pandemic, since the beginning of 2020, cancer patients across Europe have had an even tougher fight due to the overload of medical services and measures to restrict access to medical care. In the case of many patients, the result of postponing care has determined more complex treatments, worsening quality of life and reducing the chance of survival.

The aim of our study was to investigate the impact of delayed diagnosis on the staging of digestive cancers within the context of health crisis management while considering bioethical principles. This is a crucial area of research, especially in the context of health crises where healthcare systems may face challenges and disruptions.

## 2. Materials and Methods

### 2.1. Ethics Statement

All patients gave their informed consent for inclusion before they participated in this study. This study was conducted in accordance with the Declaration of Helsinki and the protocol was approved by the Research Ethics Committee of the Grigore T. Popa University of Medicine and Pharmacy of Iasi, Romania, Approval for Doctoral Research Series J, number 34/18.01.2021, issued for Andreea Luiza Palamaru.

The medical and personal information are anonymous and the requirement for a special informed approval was therefore waived.

### 2.2. Patients

This prospective study included patients who postponed elective endoscopic examinations during the COVID-19 pandemic. Thus, 205 patients who underwent elective endoscopic diagnostic procedures within the Institute of Gastroenterology and Hepatology, St. Spiridon Clinical Emergency Hospital of Iasi, Romania, after the lifting of sanitary restrictions, between 1 April 2021 and 1 April 2022, were included in the study.

### 2.3. Patient Selection

All the patients who performed endoscopic explorations were over 18 years old, presented clinical signs and symptoms suggestive of digestive impairment or biological data were presented that required the performance of endoscopic exploration. They signed the informed consent form prior to the procedure. We included in this study patients who had histopathological confirmation of digestive tract cancers and who received tumor staging by Computer Tomography with TNM classification. Patients without histopathological confirmation and those in whom endoscopic procedures were performed in an emergency setting were excluded from the study.

### 2.4. Data Collection

Each patient included in this study was evaluated for the identification of risk factors through anamnesis, local clinical examination and laboratory tests, according to the protocols in force. From the patients’ anamneses, we identified the following risk factors for the digestive cancers studied: smoking, alcohol consumption and increased consumption of red meat. For each patient, we gathered the following: demographic data, treatment timelines, discovery at different cancer stages and detailed tumor staging based on both pathology and radiological assessments. Such data can be valuable for understanding the characteristics of the patient population, treatment outcomes and the relationship between the variables mentioned. Our staging was based on the TNM classification in its latest update in 2016.

The anamnesis aimed to identify the duration of persistence of the upper or lower digestive tract symptoms, the medication followed by the patient at home and also the positive personal history for infection with the SARS-CoV-2 virus. In patients on chronic oral anticoagulant therapy, the treatment was discontinued prior to endoscopic exploration to maintain a safety profile in case biopsy was required. Biologically, the hemoglobin value was determined to establish the severity of the anemic syndrome and the Fecal Occult Blood Test (FOBT) was performed. Referring to the statistical estimates, we divided the patients into 2 groups according to the period corresponding to the transition of colon cancer from a lower to a higher TNM stage. Two groups of patients were obtained: group I (193 patients) was represented by patients whose onset of symptoms was more than 6 months prior to endoscopic exploration and group II (12 patients) with onset of symptoms less than 6 months prior to endoscopic evaluation. This division of patients was motivated by the intention to demonstrate the influence of the time of persistence of symptoms on the stage at which the cancer is diagnosed. The data were collected from medical records. The equipment and materials required for the intervention were a Pentax video gastroscope, model EG-290Kp; a Pentax video colonoscope, model EC-380FK2p; and biopsy probes.

### 2.5. Statistical Analysis

The information obtained was introduced into a database using the spreadsheet program Microsoft Excel 15.20. The statistical processing of the data was carried out by means of the IBS SPSS Statistics 24 program for Mac OS. SPSS is a software program widely used in research for statistical analysis. It provides a range of tools and features that enable researchers to analyze and interpret data. SPSS provides a user-friendly interface for inputting data, and users can easily import data from various sources, including Excel and other statistical software. SPSS is powerful for conducting a wide range of inferential statistical analyses. These include *t*-tests, analysis of variance (ANOVA), regression analysis, chi-square tests, correlation analysis and more. SPSS is a versatile tool that facilitates various stages of the research process, from data entry and management to advanced statistical analyses and reporting. Its user-friendly interface makes it accessible for varying levels of statistical expertise. In one of his articles, Loffing stated that SPSS Statistics has been the most commonly reported statistical software package used for data analysis in scientific journal articles for more than two decades [6].

We used the Kruskal–Wallis Test to determine if there were statistically significant differences between the hemoglobin values in the three types of diagnoses. Subsequently, we compared the data using the Pearson Chi-squared test because we wanted to determine whether our data were significantly different from what we expected.

## 3. Results

In total, 960 patients with upper or lower digestive tract symptoms underwent elective endoscopic procedures during the mentioned period. After applying the inclusion and exclusion criteria, we identified 205 patients with histopathologically confirmed upper and lower digestive tract cancers. The patients were aged between 51 and 70 years and 63.4% of them were male patients. Lower gastrointestinal (GI) endoscopy was performed in 84.4% of the patients and 15.6% were evaluated by upper GI endoscopy (Table 1).

Regarding the clinical parameters, weight loss was most frequently reported (61.5% of cases), followed by abdominal pain (56.6% of cases), altered bowel movements (55.1% of cases) and loss of appetite, observed in 50.7% of cases; other symptoms were observed in lower percentages: epigastric pain (27.3% of cases), heartburn (20.5% of cases) and dysphagia, observed only in isolation (4.9% of cases) (Figure 1).

The persistence of symptoms for a duration of 2 years was reported by 48.3% of the patients, while 32.2% presented symptoms for one year (Figure 2).

The diagnosis of colorectal cancer was confirmed in 84.4% of the patients, gastric cancer in 10.7% and esophageal cancer in 4.9%.

We investigated the presence of iron deficiency anemia by determining the hemoglobin values of patients. The mean value observed is 9.144 ± 1.8376, with a range of variation between 5.3 and 13.9 and a median value of 9.000; the average value of hemoglobin is higher in patients diagnosed with gastric cancer (10.382 ± 2.0720), being the lowest in patients with colorectal cancer (8.981 ± 1.7806). Patients with esophageal cancer have an intermediate hemoglobin value of 9.240 ± 1.2903, and the observed differences between the hemoglobin values in the three types of diagnoses are statistically significant (*p* = 0.011) (Table 2).

We searched whether there are statistically significant associations between the positive FOBT and the diagnosis of digestive cancer. Three different cancer diagnoses were recorded with the following results: positive FOBTs were observed in 60.1% of colorectal cancer cases, while only 40.0% of esophageal cancer cases and 40.9% of gastric cancer have associated positive FOBTs.

We analyzed the presence of alarm and unspecific clinical symptoms on the three types of diagnoses followed and observed statistically significant differences for most of the investigated clinical symptoms. Heartburn is most frequently associated with esophageal cancer (present in 50.0% of cases), and epigastric pain is most frequently associated with gastric cancer (63.6% of cases), being also identified in half of patients with esophageal cancer. Dysphagia is identified in half of the patients with esophageal cancer. Altered bowel movements are most frequently associated with colorectal cancer (present in 61.3% of cases). Abdominal pain is most frequently associated with colorectal cancer, being reported in 61.8% of cases. In the case of all these clinical symptoms, the increase in frequency is statistically significant (Table 3).

The correspondence between the evolutive stage of gastrointestinal cancer and the delay in diagnosis is as follows: the T1 stage was identified in 14.1% of the patients diagnosed with gastrointestinal cancer, two being classified with the T1a stage; 38% presented the T2 stage, seven being classified with the T2b stage; T3 was objectified in 35.7% of cases; and 12.2% presented stage T4.

The patients who presented digestive symptoms for more than six months were diagnosed in advanced stages of cancer compared to patients who reported a shorter period of symptoms. Thus, the delay in diagnosis correlates with a more advanced stage of gastrointestinal cancer. No patient with symptoms under six months had metastases.

Stage N0 was identified in 2% of patients, while stages N1 and N2 were found in almost equal percentages, 48.3% and 49.8%, respectively. The proportion of patients without metastases was 23.4%, while almost a quarter of patients were classified as M1 stage. Among the patients with persistent symptoms for more than 6 months, only 12.3% were classified in stage T1, with stage T2 representing 39.2% and T3 36.0%. The percentage of patients in the T4 stage (12.2%) is slightly higher than the similar patients with symptoms under 6 months (Table 4).

The Eastern Cooperative Oncology Group (ECOG) performance status is a scale used by different specialists to assess how the disease affects the patient’s daily skills and activities and to determine the appropriate treatment and prognosis of the disease. The ECOG provides information about the patient’s prognosis and can help estimate survival. Patients with higher ECOG scores may have poorer prognoses, which may influence treatment decisions and care management. Overall, the ECOG is a useful tool in the assessment of cancer patients, providing insight into their overall health status and contributing to personalized care and treatment decisions.

Regarding the performance status reported by the ECOG investigation, the distribution of patients is relatively even; almost one-third of patients (29.8%) have status 0, 23.4% are reported with status 1, almost one-third (30.2%) are reported with status 2 and the fewest cases (16.6%) are reported with performance status 3.

The ECOG performance status is also statistically significantly associated with the presence of a positive FOBT (*p* = 0.001); thus, the positive FOBT was mainly observed in patients with ECOG performance status 3 (82.4% of them). In the other categories of patients, the percentages with a positive FOBT are lower: 51.6% of those with an ECOG 2 status, 64.6% of those with an ECOG 1 status and only 42.6% of those with an ECOG 0 status (Table 5).

Statistically significant gender differences are observed in terms of the ECOG performance status (*p* = 0.001). Thus, among patients with status 0, the vast majority are men (80.3%), with the proportion of men decreasing significantly between patients with status 1 (68.8%) and those with status 2 or 3 (50.0%), respectively (Table 6).

## 4. Discussion

The COVID-19 pandemic had an undeniable effect on the health system in Romania and worldwide with an estimated 2.3 million cancer surgery procedures canceled during the height of the pandemic [7,8,9,10]. Serious concerns were related to medical errors, secondary to anxiety and burnout [7]. Thus, in both Europe and the United States of America, a large number of gastrointestinal cancers reportedly remained undiagnosed or untreated because patients with alarm or unspecific symptoms either postponed endoscopic investigations for fear of infection with the SARS-CoV-2 virus or did not have access to these examinations due to health policies imposed by the WHO within the whole of Europe [11,12,13].

Similar to other European countries, we found that Romanian patients needed to postpone endoscopic procedures despite so-called red flag signs that would have required diagnostic procedures. Kapoor et al. evaluated the diagnostic accuracy of alarm symptoms in a clinical prediction model for cancer and prospectively used this model in a cohort study. Their study showed that dysphagia and weight loss significantly were predictive factors for digestive cancer. Furthermore, the most common alarm symptom reported by patients in our study was weight loss, followed by abdominal pain [14]. In a recent cohort study, Rasmussen et al. evaluated the prevalence of symptom experience in the general population related to specific and non-specific symptoms suggestive of colorectal cancer. Persistent abdominal pain was reported as the most common specific alarm symptom [15].

Among the 960 patients that underwent elective endoscopic procedures in our hospital, 21.35% have been diagnosed with digestive tract cancers. Colorectal cancer was most frequently diagnosed.

Therefore, our data showed us that during the 6 months of the pandemic (1 March 2020–1 September 2020), only 202 endoscopic examinations were performed compared to 797 performed during the corresponding period in 2019. Another study driven in our center showed a dramatic decrease in diagnostic procedures, while the number of therapeutic—especially biliopancreatic procedures—remained almost the same. The number of patients diagnosed with colorectal cancer as a result of screening decreased from 33% to 5% for the March–June period in 2020 compared to the same period in 2019. The COVID-19 pandemic has had a significant impact on cancer patients’ access to care. A large number of patients with COVID-19 has overwhelmed health systems worldwide, disrupting the routine treatment of cancer patients. According to the World Health Organization (WHO), 55% of countries reported dysfunctional oncology services due to the pandemic. In many European countries, there are evident disturbances in the screening, diagnosis, treatment and long-term follow-up of cancer patients.

Hamarneh et al. showed in a recent study assessing risk factors for colorectal cancer following a positive fecal immunochemical test that iron deficiency anemia was one of the predictive factors of colorectal cancer and small intestinal cancer [16]. In a population-based cohort study, Ioannou et al. reported that among men and postmenopausal women, gastrointestinal malignancy is significantly more common in those with iron deficiency anemia [17]. In our study, all 205 patients diagnosed with digestive cancer were aged between 51 and 70 years. There were no premenopausal women in this study. Therefore, iron deficiency anemia is associated with an increased likelihood of gastrointestinal malignancy.

Early diagnosis and treatment have a major impact on the prognosis of any cancer [18,19] and any delay may lead to a progression of the disease and can directly influence the patient’s outcome. Subsequently, this causes a burden for the national health system. The main reason for such a burden is not only increased mortality but also the advancement of the cancer stage impacting treatment costs and outcome as some cancers may have become metastatic or inoperable during this delay. Such a phenomenon has been evaluated by several concomitant studies and has been therefore designated as stage migration, defined as a stage shift due to disease progression from the first symptoms up until reaching a positive diagnosis [20].

Given the fact that screening programs are performed with the aim of identifying resectable precancerous lesions and treatable early cancers [16], it is expected that delays in diagnosis due to the COVID-19 epidemic caused a significant burden driven by an increase in the number of preventable cancer deaths.

The FOBT is an ideal screening test for colorectal cancer because it is simple, cheap, and accepted by patients; has high sensitivity and specificity; has a good cost/efficiency ratio; and is effective in reducing morbidity and mortality from colorectal cancer. On the other hand, the test has no complications, and it is addressed to a healthy population. In the case of a positive test, it is necessary to further evaluate the patient by colonoscopy to detect or deny colonic pathology. The test is more effective if is performed periodically and repetitively (annually or every two years). Large studies have shown that repetitive use of the test in population screening decreases mortality by colorectal cancer by up to 30%.

Patients with a positive FOBT present a high risk of having undiagnosed gastrointestinal cancer and the presence of the disease correlates with an increased ECOG performance status.

Recently, an increase of over 1.5% in overall mortality related to colorectal cancer has been estimated in the UK, Canada, Australia and the Netherlands [13]. Our data showed us that during the 6 months of the pandemic (1 March 2020–1 September 2020), only 202 endoscopic examinations were performed compared to 797 performed during the corresponding period in 2019. Another study driven in our center showed a dramatic decrease in diagnostic procedures, while the number of therapeutic—especially biliopancreatic procedures—remained almost the same [21].

A study conducted by Tinmouth et al. in Canada that compared the number of colonoscopies performed from March to June 2020 with the same time period in 2019 revealed that their number decreased by 60% in 2020 compared to 2019, from 107.034 explorations in 2019 to 36.029 in 2020 [22]. Given the endoscopy suite restrictions, all the European Union patients with mild clinical symptoms chose a community hospital or nearby health center or even received treatment at home (without further examination) as most tertiary hospitals gave priority to critically ill patients. Manes et al. showed in a study carried out on the population of northern Italy a 44% decrease in the number of new diagnoses of gastrointestinal cancer, established by endoscopy with biopsy, during the period of pandemic restrictions [23].

On 10 May 2021, the European Cancer Organization (ECO) published a study whose significant conclusions are a reminder of the tough challenges faced by cancer patients and oncology services in Europe during the COVID-19 pandemic. In our study, for each of the three types of digestive cancer, we identified the alarm or unspecific symptoms that require endoscopic procedures. The benefit of endoscopic exploration is justified by the fact that these patients are associated with a risk of digestive cancer. We found abdominal pain, altered bowel habit, dysphagia, iron deficiency anemia as alarm signs and symptoms and a loss of appetite, weight loss and heartburn as unspecific symptoms. Alarm features are symptoms associated with serious gastrointestinal disease, such as neoplasm. The current guideline recommendation is that endoscopic evaluation of the high-risk patient should be based on age and alarm symptoms. Iron deficiency anemia is a problem commonly encountered in clinical practice, and the prevalence of underlying gastrointestinal cancer in iron deficiency anemia is the primary justification for urgent investigation. Our findings are similar to those presented by other studies in the literature. We found a relationship between the presence of iron deficiency anemia and the positive fecal immunochemical test, which was positive in 57.1% of the patients.

Regarding the delay in diagnosis, a recent systematic review presents the Andersen Model of Total Patient Delay and its application in cancer diagnosis. This model highlights the importance of motivation for delaying patient assessment, following three steps: A behavioral delay can be explained by the fear of infection with SARS-CoV-2 virus in the hospital, and the scheduling delay can be demonstrated by the restrictive measures adopted by the WHO in order to prevent the COVID-19 disease. The last step, treatment delay, can be associated with the difficulty of getting a hospital appointment [24].

The progression of cancer up until the time of diagnosis meant that severely narrowed the window of opportunity to a curative surgical treatment. Sud et al. emphasized the negative impact of the delay in the diagnosis of digestive cancer. In a study carried out in Great Britain in 2020, it is highlighted that a 3 to 6 month delay in cancer surgery, especially for stage 2 and 3 cancers, can have a substantial impact on survival [25].

The results are similar to those of our study in which we analyzed the correspondence between the evolutive stage of gastrointestinal cancer and the delay in diagnosis. Thus, the patients with persistent digestive symptoms were diagnosed in advanced stages of gastrointestinal cancer: 39.2% in the T2 stage and 36% in the T3 stage, while only 12.3% of the patients were caught in the T1 stage. Moreover, even after the resumption of standard activity in the endoscopy laboratory, the addressability of patients for endoscopic examinations did not exceed that of the pre-COVID-19 years, leading to an added case load burden [26].

We have also addressed several ethical management dilemmas, such as the balance between the need for prioritization and the impossibility of treating all patients equally.

It was discussed which patients should benefit from access to endoscopic explorations when the demand exceeds the ability to perform procedures (medical personnel in isolation, limited protective equipment and the risk of infection with the SARS-CoV-2 virus). The objective identification of the situation that justifies the restrictions of access to endoscopy is necessary. It is crucial to acknowledge the potential ethical dilemmas that may arise with such an approach [27]. Thus, a utilitarian approach to the lockdown question may be prepared to override the right to privacy or liberty to protect well-being.

There were multifaceted challenges that the COVID-19 pandemic posed to healthcare systems globally, with a particular emphasis on its impact on Low- and Middle-Income Countries (LMICs) [28,29].

Improving cancer management in challenging times requires a multifaceted approach that addresses different aspects of patient care, communication and healthcare infrastructure. In Romania, the situation of cancer patients is much worse than in most EU countries, as we are faced with inequalities and inequities in accessing medical services, with the lack of resources and the crisis of cytostatic drugs, the shortage of personnel and the failure of screening programs. During the pandemic, private services remained open to patients, receiving a large part of patients who could not access the public care system. A solution for cancer patients in a difficult period like the COVID-19 pandemic could be better financing private services so that more cancer patients can access them. Telemedicine enables remote consultations for cancer patients, reducing the need for in-person visits. Implementing remote monitoring technologies to track patients’ vital signs and treatment side effects can provide timely interventions. Promoting interdisciplinary collaboration among healthcare professionals, including oncologists, primary care physicians, nurses, mental health specialists and social workers, to provide holistic care is very beneficial for cancer patients. The implementation of protocols requires collaboration between health institutions, policy makers, researchers and the community. Regular evaluations and updates of the plan based on lessons learned during difficult times will contribute to the continuous improvement in cancer management.

It is essential to recognize that these changes in the work style of doctors are multifaceted, and their impact on decision making and treatment flexibility can vary based on individual circumstances and healthcare settings. Continuous efforts to address these challenges and strike a balance between efficiency and personalized care are crucial for maintaining the quality of healthcare delivery [30]. Limited access to trusted healthcare providers has led to increased anxiety and stress among patients [31]. Striking the right balance between autonomy, guidelines and distributive justice is essential for an effective and ethical response to healthcare challenges during a pandemic [32]. 

## 5. Conclusions

This is the first study assessing the post-pandemic burden of COVID-19-related restrictions in the management of digestive tract cancers in Romania. We searched whether pandemic restrictions had a direct impact on the post-pandemic healthcare burden driven by stage migration and the shifts in the morbidity and mortality of digestive tract cancers. Thus, we found that early detection of gastrointestinal malignancies has been severely affected during the pandemic restrictions. This had a direct effect on tumor stage and ECOG status progression. The study illustrates furthermore the impact of deontological bias in favor of utilitarianism and the maximization of the collective good taking precedence over the good of a narrow population group in need of an early diagnosis. Despite the fact that the pandemic is officially over, new cases of COVID-19 are diagnosed every day all over the world, so further research is needed in order to properly address such a burden.

## Figures and Tables

**Figure 1 healthcare-12-00691-f001:**
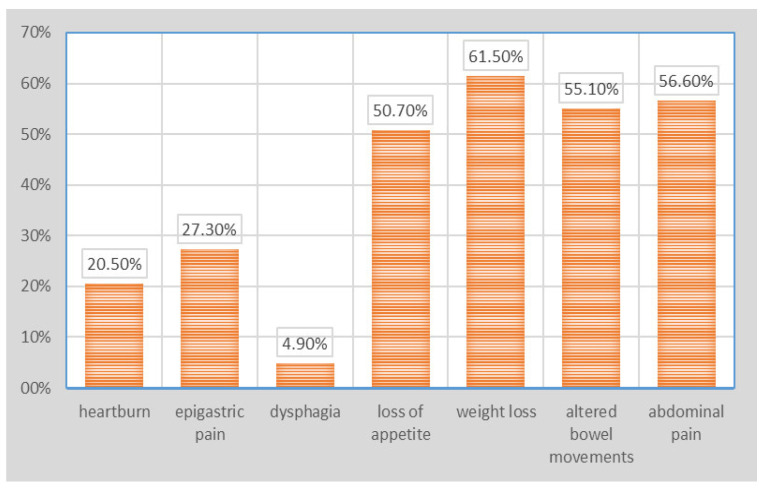
Patient distribution according to main symptoms.

**Figure 2 healthcare-12-00691-f002:**
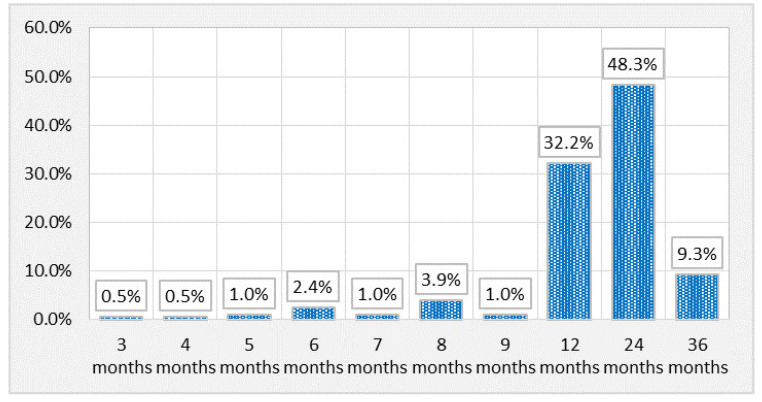
Patient distribution according to duration of symptoms.

**Table 1 healthcare-12-00691-t001:** General clinical parameters of included patients.

Endoscopic Procedures	N = 205
Lower GI endoscopy	173 (84.4%)
Upper GI endoscopy	32 (15.6%)
Diagnosis	
Colorectal cancer	173 (84.4%)
Esophageal cancer	10 (4.9%)
Gastric cancer	22 (10.7%)
T stage	
T1	29 (14.1%)
T2	78 (38%)
T3a	23 (11.3%)
T3b	50 (24.4%)
T4	25 (12.2%)
ECOG	
0	61 (29.8%)
1	48 (23.4%)
2	62 (30.2%)
3	34 (16.6%)

**Table 2 healthcare-12-00691-t002:** Comparative assessment of hemoglobin values and the 3 types of diagnoses.

Hemoglobin		N	Average Value	Standard Error of the Mean	Standard Deviation SD	Min	Max	Median	Kruskal–Wallis Test	
									H	*p*
Total		205	9.144	0.1283	1.8376	5.3	13.9	9.000		
Diagnostic	Colorectal cancer	173	8.981	0.1354	1.7806	5.3	13.2	8.800	9.008	0.011
	Esophageal cancer	10	9.240	0.4080	1.2903	7.2	11.1	9.050		
	Gastric cancer	22	10.382	0.4417	2.0720	6.9	13.9	10.350		

**Table 3 healthcare-12-00691-t003:** Correlations between symptoms and the positive diagnoses.

		Diagnostic						Pearson Chi-Squared Test
		Colorectal Cancer		Esophageal Cancer		Gastric Cancer		
		N	%	N	%	N	%	
Heartburn	Yes	31	17.9%	5	50.0%	6	27.3%	Chi-square = 6.669
	No	142	82.1%	5	50.0%	16	72.7%	*p* = 0.036
Epigastric pain	Yes	37	21.4%	5	50.0%	14	63.6%	Chi-square = 20.271
	No	136	78.6%	5	50.0%	8	36.4%	*p* = 0.000
Dysphagia	Yes	4	2.3%	5	50.0%	1	4.5%	Chi-square = 46.338
	No	169	97.7%	5	50.0%	21	95.5%	*p* = 0.0000
Loss of appetite	Yes	85	49.1%	6	60.0%	13	59.1%	Chi-square = 1.136
	No	88	50.9%	4	40.0%	9	40.9%	*p* = 0.567
Weight loss	Yes	105	60.7%	6	60.0%	15	68.2%	Chi-square = 0.472
	No	68	39.3%	4	40.0%	7	31.8%	*p* = 0.790
Altered bowel movements	Yes	106	61.3%	1	10.0%	6	27.3%	Chi-square = 17.773
	No	67	38.7%	9	90.0%	16	72.7%	*p* = 0.000
Abdominal pain	Yes	107	61.8%	2	20.0%	7	31.8%	Chi-square = 12.893
	No	66	38.2%	8	80.0%	15	68.2%	*p* = 0.002
Total		173	100.0%	10	100.0%	22	100.0%	

**Table 4 healthcare-12-00691-t004:** Comparative assessment of the delay in positive diagnosis according to T stage.

Chi-Square Test Chi^2^ = 11.090 *p* = 0.197		The Delay in Diagnosis				Total	
		Under 6 Months		More than 6 Months			
		N	%	N	%	N	%
T Stage	T1	5	45.5%	22	11.3%	27	13.2%
	T1a	0	0.0%	2	1.0%	2	1.0%
	T2	2	18.2%	69	35.6%	73	34.6%
	T2b	0	0.0%	7	3.6%	7	3.4%
	T3a	1	9.1%	22	11.3%	23	11.2%
	T3b	2	18.2%	48	24.7%	51	24.4%
	T4	0	0.0%	1	0.5%	1	0.5%
	T4a	1	9.1%	20	10.3%	21	10.2%

**Table 5 healthcare-12-00691-t005:** Comparative values of the FOBT according to ECOG performance status.

		ECOG Performance Status								Pearson Chi-Squared Test
		0		1		2		3		
		N	%	N	%	N	%	N	%	
FOBT	negative	35	57.4%	17	35.4%	30	48.4%	6	17.6%	Chi-square = 15.927
	positive	26	42.6%	31	64.6%	32	51.6%	28	82.4%	*p* = 0.001
Total		61	100.0%	48	100.0%	62	100.0%	34	100.0%	

**Table 6 healthcare-12-00691-t006:** Distribution of patients by demographic indicators, compared according to ECOG performance status.

		ECOG Performance Status								Pearson Chi-Squared Test
		0		1		2		3		
		N	%	N	%	N	%	N	%	
Sex	M	49	80.3%	33	68.8%	31	50.0%	17	50.0%	Chi-square = 15.556
	F	12	19.7%	15	31.3%	31	50.0%	17	50.0%	*p* = 0.001
	40–60 years	11	18.0%	8	16.7%	14	22.6%	13	38.2%	Chi-square = 6.475

## Data Availability

The data presented were included in this study.

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
