# Peer review of "Post-Pandemic Burden of COVID-19-Related Restrictions in the Management of Digestive Tract Cancers: A Single Center Study"

_healthcare, 2024, doi:10.3390/healthcare12060691_

Round 1
Reviewer 1 Report
Comments and Suggestions for Authors
Overall:
The authors examine the details of cases in which endoscopy was performed after the COVID-19 pandemic (April 1st, 2021 - April 1st, 2022) and histologically confirmed as gastrointestinal cancer. The authors state that early detection of gastrointestinal malignancies was greatly affected during the pandemic regulations. However, the authors do not compare their data to pre-pandemic data, and the data are not interpreted in the discussion.
Comments:
1. Please compare with pre-pandemic data where possible.
2. Please explain the interpretation of what is described in the results in the discussion.
3. The descriptions in Tables and Figures are not clear, please add or correct them.
4. Table1: The total number of T stage cases is not 205.
5. Please explain the necessity of fecal occult blood data in this article.
6. Please explain the need for data on the relationship between ECOG performance status, fecal occult blood, and gender in this article.
Author Response
Response to Reviewer 1 Comments
|
||
1. Summary |
|
|
Thank you very much for taking the time to review this manuscript. Please find the detailed responses below and the corresponding revisions/corrections highlighted/in track changes in the re-submitted files.
|
|
Comments 1: Please compare with pre-pandemic data where possible. Response 1: Thank you for pointing this out. Therefore, our data showed us that during the 6 months of the pandemic (March 1, 2020 – September 1, 2020), only 202 endoscopic examinations have been performed compared to 797 performed during the corresponding period of 2019. Another study driven in our center showed a dramatic decrease in diagnostic procedures while the number of therapeutic – especially biliopancreatic procedures remained almost the same. The number of patients diagnosed with colorectal cancer as a result of screening decreased from 33% to 5% for the March - June period of 2020 compared to the same period of 2019. The COVID-19 pandemic has had a significant impact on cancer patients' access to care. A large number of patients with COVID-19 has overwhelmed health systems worldwide, disrupting the routine treatment of cancer patients. According to the World Health Organization (WHO), 55% of countries reported dysfunctional oncology services due to the pandemic. In many European countries, there are evident disturbances in the screening, diagnosis, treatment and long-term follow-up of cancer patients. (Page number 8, paragraph 5).
Comments 2. Please explain the interpretation of what is described in the results in the discussion. Response 2: On May 10, 2021, the European Cancer Organization (ECO) published a study whose significant conclusions are a reminder of the tough challenges faced by cancer patients and oncology services in Europe during the COVID- In our study for each of the three types of digestive cancer, we identified the alarm or unspecific symptoms that require endoscopic procedures. The benefit of endoscopic exploration is justified by the fact that these patients associate a risk of digestive cancer. We found as alarm signs and symptoms the abdominal pain, altered bowel habit, dysphagia, iron deficiency anaemia and as unspecific symptoms: loss appetite, weight loss and heartburn. Alarm features are symptoms associated with serious gastrointestinal disease such as neoplasm. The current guideline recommendation is that endoscopic evaluation of the high-risk patient should be based on age and alarm symptoms. Iron deficiency anaemia is a problem commonly encountered in clinical practice, and the prevalence of underlying gastrointestinal cancer in iron deficiency anaemia is the primary justification for urgent investigation. Our findings are similar to those presented by other studies in the literature. We found a relationship between the presence of iron deficiency anaemia and the positive fecal immunochemical test, which was positive in 57.1% of the patients. (This change can be found in the text in page number 10, paragraph 2).
Comments 3: The descriptions in Tables and Figures are not clear, please add or correct them. Response 3: I made some changes to the descriptions in Tables and Figures.
Comments 4: Table1: The total number of T stage cases is not 205. Response 4: In Table 1 the T1 stage was identified in 29 patients, T2 stage in 78 patients, T3 stage in 73 patients and 25 patients presented stage T4. (Page number 4, Table 1)
Comments 5: Please explain the necessity of fecal occult blood data in this article. Response 5: Thank you for pointing this out. FOBT (Fecal Occult Blood Test) is an ideal screening test for colorectal cancer because it is simple, cheap, accepted by patients, has high sensitivity and specificity, has a good cost/efficiency ratio and is effective in reducing morbidity and mortality from colorectal cancer. On the other hand part, the test has no complications, it is addressed to a population healthy. In the case of a positive test, it is necessary to further evaluate the patient by colonoscopy, to detect or deny a colonic pathology. The test is more effective if is performed periodically, repetitively (annually or every two years). Large studies have shown that repetitive use of the test in population screening decreases mortality by colorectal cancer by up to 30%. (Page number 9, paragraph 5).
Comments 6: Please explain the need for data on the relationship between ECOG performance status, fecal occult blood and gender in this article. Response 6: Patients with a positive FOBT present a high risk of having undiagnosed gastrointestinal cancer and the presence of the disease correlates with an increased ECOG performance status. (Page number 9, paragraph 6).
|
Reviewer 2 Report
Comments and Suggestions for Authors
INTRODUCTION
1. How did the WHO impact the care of cancer patients in this sentence “Globally, the health policies imposed by the WHO have generated the impossibility of treating all patients, which has determined an ethical dilemma pertaining to the balance of utilitarianism versus deontology, regarding the patient's access to public health services”
MATERIALS AND METHODS
1. A reference for SPSS must be added.
2. The number of patients in each group must be added.
3. The risk factors used in the study must be listed.
RESULTS & DISCUSSION
1. I believe this study is more of a colorectal cancer study 84.4% of patients are diagnosed with CRC authors can’t use only 15.6% of the subjects with different cancer types and oriented their study as a gastrointestinal study.
2. The clinical parameters are different between the cancer types included in the study a subsequent analysis of the cohort by cancer type will be better for CRC (CC V.s RC), OC, and GC separately.
DISCUSSION
The pandemic has decreased the healthcare system not only in low and medium-income cantoris but also in well-developed cantoris and most patients were influenced by these changes especially cancer patients who need rapid diagnosis and disease management. The authors should propose a better plan or protocols to improve cancer management during a difficult time. The discussion can be improved.
CONCLUSION
Overestimated is not related to the results.
Comments on the Quality of English LanguageEnglish of the study is acceptable
Author Response
|
|||
1. Summary |
|
|
|
Thank you very much for taking the time to review this manuscript. Please find the detailed responses below and the corresponding revisions/corrections highlighted/in track changes in the re-submitted files.
|
Comments 1: How did the WHO impact the care of cancer patients in this sentence “Globally, the health policies imposed by the WHO have generated the impossibility of treating all patients, which has determined an ethical dilemma pertaining to the balance of utilitarianism versus deontology, regarding the patient's access to public health services”
Response 1: Thank you for pointing this out. In 2020, the year of the COVID-19 pandemic, there were numerous restrictions and legislative regulations that affected the diagnosis and treatment of cancers. Access to investigations and treatments was restricted by the policies imposed by the WHO Since the outbreak of the COVID-19 pandemic, since the beginning of 2020, cancer patients across Europe have had an even tougher fight due to the overload of medical services and measures to restrict access to medical care. In the case of many patients, the result of postponing care has determined more complex treatments, worsening quality of life and reducing the chance of survival. (Page number 2, paragraph 2)
Comments 2: A reference for SPSS must be added.
Response 2: SPSS (Statistical Package for the Social Sciences) is a software program widely used in research for statistical analysis. It provides a range of tools and features that enable researchers to analyze and interpret data. SPSS provides a user-friendly interface for inputting data, and users can easily import data from various sources, including Excel and other statistical software. SPSS is powerful for conducting a wide range of inferential statistical analyses. These include t-tests, analysis of variance (ANOVA), regression analysis, chi-square tests, correlation analysis, and more. SPSS is a versatile tool that facilitates various stages of the research process, from data entry and management to advanced statistical analyses and reporting. Its user-friendly interface makes it accessible for varying levels of statistical expertise. (Page number 3, paragraph 5)
Comments 3: The number of patients in each group must be added.
Response 3: We divided the patients into 2 groups according to the period corresponding to the transition of colon cancer from a TNM stage to an advanced one. 2 groups of patients were obtained: group I (193 patients) was represented by patients whose onset of symptoms was more than 6 months prior to endoscopic exploration and group II (12 patients) with onset of symptoms less than 6 months prior to endoscopic evaluation. (Page number 3, paragraph 3)
Comments 4: The risk factors used in the study must be listed.
Response 4: Thank you for pointing this out. From the patients' anamnesis, we identified the following risk factors for the digestive cancers studied: smoking, alcohol consumption and increased consumption of red meat. (Page number 3, paragraph 2)
Comments 5: I believe this study is more of a colorectal cancer study 84.4% of patients are diagnosed with CRC authors can’t use only 15.6% of the subjects with different cancer types and oriented their study as a gastrointestinal study.
Response 5: We agree with this comment. When we started the research, we did not expect to identify colorectal cancer in such a significant percentage.
Comments 6: The clinical parameters are different between the cancer types included in the study a subsequent analysis of the cohort by cancer type will be better for CRC (CC V.s RC), OC, and GC separately.
Response 6: We agree with this comment. For each type of digestive cancer, we took into account the dominant symptomatology, but in the future we will also take into account a separate analysis. (Page number 3, paragraph 2)
Comments 7: The pandemic has decreased the healthcare system not only in low and medium-income cantoris but also in well-developed cantoris and most patients were influenced by these changes especially cancer patients who need rapid diagnosis and disease management. The authors should propose a better plan or protocols to improve cancer management during a difficult time. The discussion can be improved.
Response 7: Thank you for pointing this out. Improving cancer management in challenging times requires a multifaceted approach that addresses different aspects of patient care, communication, and healthcare infrastructure. In Romania, the situation of cancer patients is much worse than in most EU countries, as we are faced with inequalities and inequities in accessing medical services, with the lack of resources and the crisis of cytostatic drugs, the shortage of personnel and the failure of screening programs. During the pandemic, private services remained open to patients, receiving a large part of patients who could not access the public care system. A solution for cancer patients in a difficult period like the one of the COVID-19 pandemic could be a better financing of private services, so that more cancer patients can access them. Telemedicine enables remote consultations for cancer patients, reducing the need for in-person visits. Implementing remote monitoring technologies to track patients' vital signs and treatment side effects can provide timely interventions. Promoting interdisciplinary collaboration among health care professionals, including oncologists, primary care physicians, nurses, mental health specialists, and social workers, to provide holistic care is very beneficial for cancer patients. Implementation of protocols requires collaboration between health institutions, policy makers, researchers and the community. Regular evaluations and updates of the plan based on lessons learned during difficult times will contribute to the continuous improvement of cancer management. (Page number 11, paragraph 3)
Comments 8: Overestimated is not related to the results.
Response 8: Thank you for pointing this out. I will take into account your opinion.
Reviewer 3 Report
Comments and Suggestions for Authors
This is a relevant analysis of the impact of COVID-19 on timely detection and treatment of gastrointestinal cancer. The message should be sharpened by (1) focusing more on this issue, and (2) substantial shortening of the general discussion of ethics and medical welfare in the world. Symptoms should be reported with a view to their role in early cancer detection, with emphasis on how they could be used to select patients for early attention.
In sum, instead of dwelling much on deontology and utilitarianism, the authors could use their experience to make recommendations on which symptoms and signs should prompt immediate action.
Recommended shortening:
L37 to l44: Deletable
L306 to l313: Deletable
l290 to l305: is straying from the central message. The dilemma should be addressed, but this paragraph presents a number of issues that are already well-known without adding much and would benefit from shortening.
Details:
L108: “TNM stage” is not the opposite of “advanced stage”. Also, all patients were divided in groups 1 and 2, not just those with colon cancer. Indeed, the groups where defined by duration of symptoms before endoscopy, not by the mentioned transition. Also, to identify the mentioned transition “from a TNM stage to an advanced one” would have required more than one endoscopy.
L163: The low numbers of esophageal and cancer patients practically preclude a meaningful comparison between positive and negative FOBT. The statement” these differences, although present, do not exceed the threshold of statistical significance ” is a bit doubtful, as it is difficult to state that there is a difference without statistical significance. You may call it a nominal difference that is lacking statistical significance. There is no real need to report this in a table.
L178: It is unclear, how the correspondence between cancer stage and delay in diagnosis was found. The paragraph just reports various tumor stages (some of them with an unclear notation: ”…2 (%??) being classified …- 7 (???) being classified…”) with percentages that do not seem to add up. This paragraph is central for the topic of the manuscript and should be presented in more detail. A table would probably be helpful.
L186: Again, this is the central topic. The main conclusion would be better supported in a figure, with no need to report the totals (last two columns). Of note, cases add up to 202, while elsewhere it is 205 throughout.
L238: Complicated style. Please cite this study in a more precise way: certainly not all patients enrolled in reference [16] showed cancer and the association with iron deficiency was not valid for premenopausal women.
L276: Would you say that any progression closed the window of opportunity? Maybe a wording like “severely narrowed the window of opportunity” would be more to the point.
L282: Again, this is the central message, which calls for a figure (see comment to l178).
L298: “…and also…” – A break in the sentence that ends up as a difficult construct; please reword.
Language and typos
“ECOG”: Define on first usage
Abstract,l 23: have had
L80: Hepatology
L84: complicated construction of sentence
L158: complicated construction of sentence
L169: pain is most
L175: Proposal for a more precise description: “In the case of all these clinical symptoms, the increase in frequency is statistically significant (Table 4).”
L220: Similarly
L319: have led
Figure 2: replace “luni” with “months”
Table 2: Reporting means, SEM, SD, Min, Max, and Median in the table seems be a bit overdone. Actually, all relevant values are already indicated in the text. After adding the statistical test to that paragraph the table can be removed.
Table 3: There is no real need to have this table.
Table 4: Here and elsewhere leading zeroes should not be omitted, as, for example, in “p= .036”.
Table 5: replace “patrat” with “square”
Table 6: missing number (FOBT negative, first column)
Table 7: replace “ani” with “years”
Comments on the Quality of English LanguageBetter than some other papers, but needs editing. I have made some suggestions, but not seldom the reader is losing track in misconstructed sentences.
Author Response
|
|||
1. Summary |
|
|
|
Thank you very much for taking the time to review this manuscript. Please find the detailed responses below and the corresponding revisions/corrections highlighted/in track changes in the re-submitted files.
|
L 108: The TNM stage was determined by performing the histopathological diagnosis
“ECOG”: Define on first usage
Response: Thank you for pointing this out. The ECOG performance status is a scale used by different specialists to assess how the disease affects the patient's daily skills and activities and to determine the appropriate treatment and prognosis of the disease. ECOG provides information about the patient's prognosis and can help estimate survival. Patients with higher ECOG scores may have poorer prognoses, which may influence treatment decisions and care management. Overall, ECOG is a useful tool in the assessment of cancer patients, providing insight into their overall health status and contributing to personalized care and treatment decisions. (Page number 7, paragraph 3)
Language and typos
Response: Language changes and typos can be found marked in red in the uploaded document.
Round 2
Reviewer 1 Report
Comments and Suggestions for Authors
In response to the comments, it has been corrected, but no references have been inserted. Please add references.
Author Response
1. Summary |
|
|
Thank you very much for taking the time to review this manuscript. Please find the detailed responses below and the corresponding revisions/corrections highlighted/in track changes in the re-submitted files. I appreciate your comments. I have also added references, according to the recommendations.
|
Reviewer 2 Report
Comments and Suggestions for Authors
Thank you for taking into consideration most of my comments, your manuscript has been readily improved but there is still room for improvement.
Comments on the Quality of English Language
Acceptable
Author Response
1. Summary |
|
|
Thank you very much for taking the time to review this manuscript. Please find the detailed responses below and the corresponding revisions/corrections highlighted/in track changes in the re-submitted files. I appreciate your comments. Comments 2: A reference for SPSS must be added. Response 2: SPSS (Statistical Package for the Social Sciences) is a software program widely used in research for statistical analysis. It provides a range of tools and features that enable researchers to analyze and interpret data. SPSS provides a user-friendly interface for inputting data, and users can easily import data from various sources, including Excel and other statistical software. SPSS is powerful for conducting a wide range of inferential statistical analyses. These include t-tests, analysis of variance (ANOVA), regression analysis, chi-square tests, correlation analysis, and more. SPSS is a versatile tool that facilitates various stages of the research process, from data entry and management to advanced statistical analyses and reporting. Its user-friendly interface makes it accessible for varying levels of statistical expertise. In one of his articles, Loffing stated that SPSS Statistics, is the most commonly reported statistical software package used for data analysis in scientific journal articles since more than two decades. (Page number 3, paragraph 4)
|
Reviewer 3 Report
Comments and Suggestions for Authors
You have chosen to simply ignore most of my content-related comments, so I have abstained from further acivities on this manuscript.
Comments on the Quality of English LanguageGood enough, minor improvements may be possible; better than numerous other manuscripts
Author Response
1. Summary |
|
|
Thank you very much for taking the time to review this manuscript. Please find the detailed responses below and the corresponding revisions/corrections highlighted/in track changes in the re-submitted files.
Recommended shortening: According to the recommendations from the first round, I tried to substantial shortening of the general discussion of ethics and medical welfare in the world.
L37 – L44: In this paragraph I tried to emphasize the importance of improving the management of digestive cancers in the context of health crises.
L306 – L313: Thank you for the observation. I deleted that paragraph and replaced it with the phrase: “There was multifaceted challenges that the COVID-19 pandemic has posed to healthcare systems globally, with a particular emphasis on its impact on Low and Middle-Income Countries (LMICs)” (Page number 11, paragraph 1)
L290 – L305: I shortened it according to your recommendations and replaced it with the sentence: “It's crucial to acknowledge the potential ethical dilemmas that may arise with such an approach”. (Page number 10, paragraph 7)
Details: |
L 108: In this sentence I wanted to refer to the progression of the TNM stage. I also modified the text with the phrase ”from a lower to a higher TNM stage”.
(Page number 3, paragraph 3)
L 163: Thank you for pointing this out. Indeed, the number of esophageal cancer cases is low. I deleted the sentence “These differences, although present, do not exceed the threshold of statistical significance”.
(Page number 6, paragraph 2)
I deleted table 3.
L 178: I added this phrase to the text to highlight the correspondence between the delay in diagnosis and the duration of the symptoms' persistence “Patients who presented digestive symptoms for more than six months were diagnosed in advanced stages (T3a, T3b) of cancer, compared to patients who reported a shorter period of symptoms. Thus, the delay in diagnosis correlates with a more advanced stage of gastrointestinal cancer. No patient with symptoms under six months had metastases.”
(Page number 7, paragraph 1)
L 186: Thank you for pointing this out. The total number of patients included in the study is 205.
I carefully checked the number of patients in each category and corrected it.
(Page number 7, table 4)
L 238: Thank you for pointing this out. In a population-based cohort study, Ioannou et al. reported that none of the premenopausal women with iron deficiency were diagnosed with gastrointestinal malignancy. Among men and postmenopausal women, gastrointestinal malignancy is significantly more common in those with iron deficiency than in persons with normal serum iron saturation and hemoglobin levels. In our study all the 205 patients diagnosed with digestive cancer were aged between 51 and 70 years. There were no premenopausal women in the study. Therefore, iron deficiency anemia, is associated with an increased likelihood of gastrointestinal malignancy.
I rephrased in the text as you suggested to me the first time.
(Page number 9, paragraph 2)
L 276: Thank you for your opinion. Really, progression of cancer up until the time of diagnosis meant that severely narrowed the window of opportunity to a curative surgical treatment. I made the correction in the text.
(Page number 10, paragraph 4)
L 282: Thank you for the observation. Really, this is the central message: the delay in diagnosis correlates with a more advanced stage of gastrointestinal cancer. I previously showed the correspondence between the TNM stage and the number of cases.
L 298: Thank you for pointing this out. I rephrased.
Language and typos
Response: Language changes and typos can be found marked in red in the uploaded document.
“ECOG”: Define on first usage
Response: Thank you for the observation. ECOG definition has been inserted in the text.
(Page number 7, paragraph 3)
L 158: Thank you for pointing this out. I rephrased.
(Page number 6, paragraph 1)
L175: Thank you for your opinion. I replaced in the text with the phrase: “In the case of all these clinical symptoms, the increase in frequency is statistically significant”.
(Page number 6, paragraph 2)
Figure 2: I replaced “luni” with “months”
Table 2: Thank you for your opinion. I consider it important the comparative assessment of hemoglobin values and the 3 types of diagnoses.
Table 3: I deleted table 3.
Table 4: I completed according to the recommendations.
Table 5 (now Table 4): I replaced “patrat” with “square”.
Table 6 (now Table 5): I completed according to the recommendations.
Table 7 (now Table 6): I replaced “ani” with “years”.